# Transmission Dynamics of Hepatitis B: Analysis and Control

## Abstract

The infection of hepatitis B attacks the liver and can produce acute and chronic diseases, while it is a major health problem and life-threatening around the globe. The control of this infection is a difficult task due to several reasons such as variation of human behavior, proper medication, vaccination, and existence of a large number of carries, etc., but understanding the dynamics of the infection helps to design appropriate control strategies. Thus, a proper complex dynamical system is needed to find the stability conditions and propose intervention strategies for forecasting the control of hepatitis B virus transmission. We formulate a model that will be helpful to investigate the temporal dynamics and suggest control strategies for hepatitis B infection. The well-posedness of the proposed model will be shown, and used to find the threshold parameter to analyze the model equilibria and its stability. We also perform the sensitive analysis of the threshold quantity to quantify the most sensitive epidemic parameters. Based on the temporal dynamics and sensitivity, we investigate effective methods to minimize the infection of hepatitis B, and develop the algorithms to support the theoretical results with the help of numerical simulations.

## 1 Introduction

Hepatitis means liver inflammation produced by the virus's bacterial infections and continuous exposure to alcohol or drugs (Ganem & Prince, 2004). Hepatitis B infection is caused by a non-cytopathic virus called the hepatitis B virus (HBV). The viruses are in vaginal fluids, blood, and semen, and are transferred in multiple ways from one individual to other. Major routes of the transmissions are blood, sharing razors, toothbrushes, and unprotected sexual contacts (Chang, 2007). Another source of HBV transmission is maternal transmission (i.e., from an infected mother to her child). However, this virus can not be transmitted from causal contact (McMahon, 2005). The infectious hepatitis B has multiple infection phases: acute and chronic carries. The first phase refers to the initial three months after someone becomes infected with hepatitis B. The immune system can recover in this stage without taking any treatment/hospital care usually, however, may also lead to a long-term infection indicating the chronic phase for someone. The individual with chronic HBV stage often has no acute history of infection. The chronic hepatitis B phase is a severe stage and causes many complications, for example, liver scarring, failure of the liver, and liver cancer (Ringehan et al., 2017). In the chronic hepatitis B phase, there is a need for treatment with medicines, containing oral antiviral agents. The proper treatment and hospital care will be necessary for the rest of their lives, because this can slow cirrhosis progression and reduce the liver cancer incidence with an improvement in long-term survival. In 2021, WHO recommended oral treatments for the hepatitis B virus as the most potent drugs. The symptom of hepatitis B includes skin yellowing, abdominal pain, urine darkness, fever, and loss of appetite, etc. But for every individual, this is not common to suffer from hepatitis B because 40% of the acute individuals have no symptoms in the acute stage of the infection. Although the vaccine for hepatitis B is available, new cases still have been reported and the infection of hepatitis B is one of the main public health issues that produce a high mortality rate around the globe (Shepard et al., 2006). Worldwide two billion of the population are infected with hepatitis B, among which the number of chronic hepatitis B is 360 million (Libbus & Phillips, 2009). Viral hepatitis B is a leading source of death among other diseases while producing 820000 deaths in 2019, with 1.5 million newly infected.

Mathematical modeling is a useful tool and has been extensively used to test various theories related to the dynamics of infectious diseases. Various mathematicians, researchers, and biologists used mathematical models to study the communication of transmittible diseases (Rao & Kumar, 2015; Lessler et al., 2016; Ndeffo-Mbah et al., 2018; Li, 2018). The temporal dynamics of hepatitis B also have a rich literature, and a variety of epidemic models have been used frequently to know the dynamics of the disease. For example, a model has been reported to analyze the dynamics of hepatitis B in Ghana by Dontwi et al. (2014). The dynamics of the hepatitis B virus with fractional derivatives have been studied by Ullah et al. (2018a;b). Similarly, many more studies have been reported to discuss the dynamics of hepatitis B (Koonprasert et al., 2016; Khan et al., 2019). However, the recent models are aimed to analyze the temporal dynamics of the infection using simple disease transmission rate rather than probability-based transmission, which is not realistic especially for a disease having multiple infection phases. Since the contagious infection of hepatitis B has multiple phases and it would be better to use probability-based transmission in the inflow and outflow of individuals from one compartment to another compartment. We try to fill the gap by formulating a model using probability-based rate of transmission according to the characteristics of hepatitis B virus transmission. The model will be developed using the idea of classical susceptible-infected-recovered (SIR) model and probability-based transmission of individuals. We then discuss the existence of the model solution using the Cauchy abstract equation and calculate the threshold parameter of the model to analyze the sensitivity of important parameters. The threshold parameter and its sensitivity are discussed with the help of the next-generation matrix method and sensitivity index, respectively. We also investigate the qualitative behaviors of the model to derive the stability conditions using linear stability analysis. Based on sensitivity and stability conditions, we use optimal control theories to formulate effective methods for hepatitis B eradication. Finally, algorithms for the proposed model will be developed to present the numerical simulations and verify the theoretical analysis.

## 2 PRELIMINARIES

In this section, we recall some essential concepts, preliminaries, definitions, and methods for the analysis of proposed model that will be helpful for getting our results.

Let $\mathcal{R}^n$ and $\mathcal{R}^n_+$ respectively represent the space of n-tuple and the space of n-tuple with non-negative entries. The system of non-linear autonomous differential equation then looks like

$$\frac{d\vec{u}}{dt} = \psi(\vec{u}(t)), \quad \vec{u}(t_0) = \vec{u}_0, \tag{1}$$

where $\vec{u} \in \mathcal{R}^n$ and the function $\psi$ implicitly depends only on the dependent variable $\vec{u}$ and not on the independent variable $t$, for example, $\frac{du}{dt} = (a - bu(t))u(t)$. Let $J$ be the variational matrix of Eq.(1), then

$$J = \begin{bmatrix} \frac{\partial \psi_1}{\partial u_1} & \cdots & \frac{\partial \psi_n}{\partial u_n} \\ \vdots & \vdots & \vdots \\ \frac{\partial \psi_n}{\partial u_1} & \cdots & \frac{\partial \psi_n}{\partial u_n} \end{bmatrix}. \tag{2}$$

**Theorem 1** *(Yassen, 2005) Let us assume that $\mathcal{P}(x)$ is a polynomial of degree $n$, such that*

$$\mathcal{P}(x) = x^n + \alpha_1 x^{n-1} + \ldots + \alpha_n,$$

*where $\alpha_1, \alpha_2, \ldots, \alpha_n$, are constants. The $n$-Hurwitz matrices for the above polynomial are defined by*

$$\mathcal{H}_1 = \alpha_1, \quad \mathcal{H}_2 = \begin{bmatrix} \alpha_1 & 1 \\ \alpha_3 & \alpha_2 \end{bmatrix}, \quad \mathcal{H}_3 = \begin{bmatrix} \alpha_1 & 1 & 0 \\ \alpha_3 & \alpha_2 & \alpha_1 \\ \alpha_4 & \alpha_3 & \alpha_3 \end{bmatrix},$$

$$\vdots$$

$$\mathcal{H}_n = \begin{bmatrix} \alpha_1 & 1 & 0 & 0 & \ldots & 0 \\ \alpha_3 & \alpha_2 & \alpha_1 & 1 & \ldots & 0 \\ \vdots & \vdots & \vdots & \vdots & \ldots & \vdots \\ 0 & 0 & 0 & 0 & \ldots & \alpha_n \end{bmatrix}.$$

*It should be noted that for $m > n$, $\alpha_m = 0$. If the determinants of the Hurwitz matrices are positive i.e. $\det(\mathcal{H}_1), \det(\mathcal{H}_2), \ldots, \det(\mathcal{H}_n) > 0$, the roots of $\mathcal{P}(x)$ will be negative or having negative real parts.*

## 2.1 CLASSIC SIR MODEL

In the history of epidemiology of the infectious disease, a classical model for the infectious disease mitigation was presented by Kermack and McKendrick which looks like

$$\frac{d\mathcal{S}}{dt} = -\beta\mathcal{I}\mathcal{S}, \quad \frac{d\mathcal{I}}{dt} = \beta\mathcal{I}\mathcal{S} - \gamma\mathcal{I}, \quad \frac{d\mathcal{R}}{dt} = \gamma\mathcal{I},$$

where the state variables, $\mathcal{S}$, $\mathcal{I}$ and $\mathcal{R}$ describe the susceptible, infectious, and recovered/removed population, respectively, while the disease mitigation rate is $\beta$ and the recovery/removal rate is $\gamma$.

**Definition 1** *Let $\vec{u}^*$ is a fixed (equilibrium) point of the dynamical system (1) and $\psi$ is a real valued $C^1$-function defined in some neighbourhood of $\vec{u}^*$ such that*

*a. $\psi(\vec{u}^*) = 0$, and $\psi(\vec{u}) > 0$ if $\vec{u} \neq \vec{u}^*$,    b. $\dot{\psi}(\vec{u}) \leq 0$, and $\dot{\psi}(\vec{u}) = 0$ if $\vec{u} = \vec{u}^*$,*

*then the function $\psi(\vec{u})$ is Lyapunov, while the fixed point $\vec{u}^*$ is stable asymptotically.*

**Definition 2** *(Samsuzzoha et al., 2013; Ngoteya & Gyekye, 2015) The basic reproductive number is a very important quantity in the study of epidemiological models. Usually, the estimation of epidemic parameters and uncertainties affect this quantity. The sensitivity analysis describes the relation, and its relative impact between the threshold quantity and epidemic parameters is defined as $\frac{\varphi}{\mathcal{R}_0}\frac{\partial\mathcal{R}_0}{\partial\varphi}$, where $\varphi$ is any epidemic parameter and $\mathcal{R}_0$ is the threshold quantity.*

## 2.2 PONTRYAGIN'S MAXIMUM PRINCIPLE

Let $\mu = (\mu_1(t), \mu_2(t), \ldots, \mu_n(t))$ and $s = (s_1(t), \ldots, s_n(t))$ denote the state and control measures, respectively. Then the optimal problem for the dynamical system is

$$\frac{ds}{dt} = g(t, s, \mu(t)), \quad s(0) = s_0, \quad 0 \leq t \leq T \quad \text{is} \quad \min\left\{\phi(T, s(T)) + \int_0^t g(t, s, \mu)\right\},$$

then the Lagrangian $L$ and the Hamiltonian $H$ take the form

$$L(s, \mu) = g(t, s, \mu), \quad H(s, \mu, \lambda) = L(s, \mu) + \lambda f(s, \mu),$$

where

$$f(s, \mu) = (f_1(s, \mu), f_2(s, \mu), \ldots f_n(s, \mu)), \quad \lambda = (\lambda_1, \lambda_2, \ldots, \lambda_n).$$

If $(s^*, \mu^*)$ represents the optimal solution of the above problem, then a non-trivial function, denoted by $\lambda$, exists and satisfies

$$\frac{ds^*}{dt} = \frac{\partial H}{\partial\lambda}, \quad 0 = \frac{\partial H}{\partial\mu}, \quad \frac{d\lambda(t)}{dt} = -\frac{\partial H}{\partial s},$$

at $(\mu^*(t), s^*(t), \lambda(t))$, and the maximality and transversal conditions

$$H(\mu^*, s^*, \lambda) = \max_{\mu \in [0,1]} H(s^*, \mu, \lambda), \quad \text{and} \quad \lambda(T) = 0 \tag{3}$$

hold.

## 3 MODEL FORMULATION

In this section, we present the model framework and its detailed derivations. We follow the classic *SIR* model, keep in view the transfer mechanism of the disease, and have the total population, represented by $N(t)$, divided into various sub-classes. Particularly, the various epidemiological sub-classes are susceptible, acute hepatitis B, chronic/carries, recovered, and the vaccinated compartments, indicated by $\mathcal{S}$, $\mathcal{A}$, $\mathcal{C}$, $\mathcal{R}$ and $\mathcal{V}$, respectively. We assume for the model constraints as follows:

- The various parameters and groups of population, $\mathcal{S}$, $\mathcal{A}$, $\mathcal{C}$, $\mathcal{R}$, and $\mathcal{V}$ are assumed to be non-negative at time $t = 0$.
- There are two infectious phases of hepatitis B: acute and chronic, and both cause the spreading of infection while the disease transmission co-efficient for acute and chronic are respectively denoted by $\beta_a$ and $\beta_c$. Also, if $\rho$ is the probability of those who leads to the acute portion after some one infected then $(1 - \rho)$-th portion will go to the chronic phase of the infection.
- Successfully vaccinated individual will lead to the vaccinated compartment while unsuccessful will go to the susceptible group of population.
- The death induced from hepatitis B is only considered in chronically infected group of individuals.
- The inflow of maternally infected individuals go to the chronic carries.

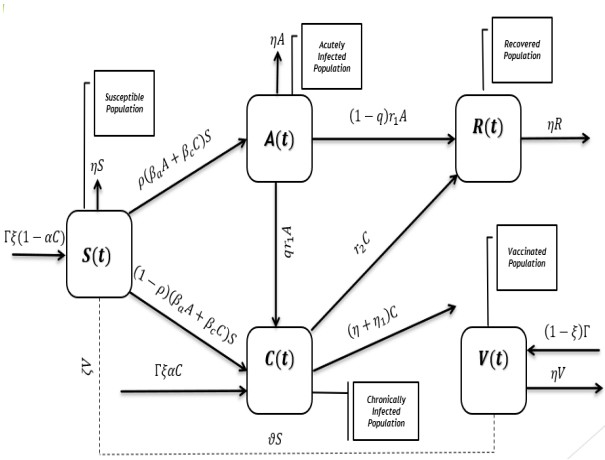

Figure 1: The graph represents the schematic process of HBV virus transmission of the proposed epidemic problem

Moreover, the schematic process of the model is depicted by Figure 1 and thus the governing equations of the epidemic problem subject to above assumptions is demonstrated by

$$
\begin{aligned}
\frac{d\mathcal{S}(t)}{dt} &= \Gamma\xi\left(1 - \alpha\mathcal{C}(t)\right) + \zeta\mathcal{V}(t) - \left(\beta_a\mathcal{A}(t) + \beta_c\mathcal{C}(t) + \vartheta + \eta\right)\mathcal{S}(t), \\
\frac{d\mathcal{A}(t)}{dt} &= \left(\beta_a\mathcal{A}(t)\mathcal{S}(t) + \beta_c\mathcal{C}(t)\mathcal{S}(t)\right)\rho - \left(r_1 + \eta\right)\mathcal{A}(t), \\
\frac{d\mathcal{C}(t)}{dt} &= \left(\beta_a\mathcal{A}(t) + \beta_c\mathcal{C}(t)\right)\mathcal{S}(t)\left(1 - \rho\right) + qr_1\mathcal{A}(t) - \left(\eta + \eta_1 + r_2 - \Gamma\xi\alpha\right)\mathcal{C}(t), \\
\frac{d\mathcal{R}(t)}{dt} &= (1 - q)r_1\mathcal{A}(t) + r_2\mathcal{C}(t) - \eta\mathcal{R}(t), \qquad \frac{d\mathcal{V}(t)}{dt} = (1 - \xi)\,\Gamma + \vartheta\mathcal{S}(t) - (\eta + \zeta)\mathcal{V}(t),
\end{aligned}
\tag{4}
$$

with initial population sizes

$$
\mathcal{S}(0), \quad \mathcal{V}(0) > 0, \quad \mathcal{A}(0), \quad \mathcal{C}(0), \quad \mathcal{R}(0) \geq 0.
\tag{5}
$$

In the above epidemic problem (4)-(5) the description of model parameters are as: $\Gamma$ is the birth rate while the hepatitis B transmission rates are $\beta_a$ and $\beta_c$. Moreover, $\eta$ is assumed to be the natural death rate. $r_1$ and $r_2$ respectively show the natural recovery rate, and recovery due to treatment. $\eta_1$ is the death rate from disease while $\vartheta$ is the vaccination rate of susceptible population. Further, the maternally infection rate (vertical transmission) is denoted by $\alpha$, and $\xi$ is the proportion of unsuccessful vaccination rate. The probability of those individuals which have no acute history while directly leading to the chronic stage is denoted by $\rho$. Similarly, $q$ is the probability of those which recover directly in acute stage.

It is worthy to mention that the proposed epidemic model is a population dynamical problem, and to show the well-possedness we describe the following propositions (all the missing proofs are given in appendix).

**Proposition 1** *The set $\mathcal{R}_+^5$ is invariant positively for the system (4).*

**Proposition 2** *The solution of the proposed epidemic problem (4) is positive subject to the initial conditions (5).*

## 4 STEADY STATES ANALYSIS

We perform a qualitative analysis of the epidemic problem. We find the model equilibria and threshold parameter to analyze the temporal dynamics of the model. We calculate the basic reproductive number to derive some sufficient conditions for the local and global dynamics of the model using linearization, Hurwitz criteria, and Lyapunov function theory.

### 4.1 STABILITY ANALYSIS

To perform the dynamical analysis of the epidemic problem, we first find the disease-free equilibrium of the system (4). Let $\mathcal{D}_f$ be the disease-free state of the proposed system described by $\mathcal{D}_f = (\mathcal{S}_f, \mathcal{A}_f, \mathcal{B}_f, \mathcal{R}_f, \mathcal{V}_f)$, where $\mathcal{S}_f = \frac{\Gamma\xi}{\eta+\vartheta}$, $\mathcal{A}_f = \mathcal{C}_f = \mathcal{R}_f = 0$, and $\mathcal{V}_f = \frac{\Gamma(1-\xi)(\eta+\vartheta)+\vartheta\Gamma\xi}{(\eta+\zeta)(\eta+\vartheta)}$. Further, the reproductive parameter, known as the threshold parameter and generally denoted by $\mathcal{R}_0$, represents the average of secondary infectious produced by an infective whenever put into susceptible individuals. Since, this quantity is the average of secondary infectives, the infection dies out if $\mathcal{R}_0 < 1$, and spreads whenever $\mathcal{R}_0 > 1$. To calculate this quantity we follow (Perasso, 2018), and let us assume that $\mathcal{Y} = (\mathcal{A}, \mathcal{C})^\top$ which implies that $\frac{d\mathcal{Y}}{dt}|_{D_f} = \mathcal{F} - \mathcal{V}$, where $\mathcal{F}$ and $\mathcal{V}$ are the variational matrices around the disease-free state $D_f$ and becomes

$$\mathcal{F} = \left[ \begin{array}{cc} \rho\beta_a\mathcal{S}_f & \rho\beta_c\mathcal{S}_f \\ (1-\rho)\,\beta_a\mathcal{S}_f & (1-\rho)\,\beta_c\mathcal{S}_f \end{array} \right], \quad \mathcal{V} = \left[ \begin{array}{cc} \eta+r_1 & 0 \\ -qr_1 & \eta+\eta_1+r_2-\Gamma\xi\alpha \end{array} \right].$$

Calculating the spectral radius of $\mathcal{F}\mathcal{V}^{-1}$, i.e., $\sigma(\mathcal{F}\mathcal{V}^{-1})$, we have that $\sigma(\mathcal{F}\mathcal{V}^{-1}) = \mathcal{R}_0$ and

$$\mathcal{R}_0 = \mathcal{R}_1 + \mathcal{R}_2 + \mathcal{R}_3, \quad \mathcal{R}_1 = \frac{\beta_c(1-\rho)\mathcal{S}_f}{(\eta+\eta_1+r_2-\Gamma\xi\alpha)}, \quad \mathcal{R}_2 = \frac{\beta_a\rho\mathcal{S}_f}{(\eta+r_1)},$$

$$\mathcal{R}_3 = \frac{\beta_cqr_1\rho\mathcal{S}_f}{(\eta+r_1)(\eta+\eta_1+\gamma_2-\Gamma\xi\alpha)}. \tag{6}$$

We now calculate the endemic state of the model. To make our calculation easier, we substitute $q_1 = \eta + \vartheta$, $q_2 = \eta + r_1$, $q_3 = \eta + \eta_1 + r_2 - \Gamma\xi\alpha$, and $q_4 = \eta + \zeta$. The disease endemic state of the model is denoted by $\mathcal{D}_e = (\mathcal{S}_e, \mathcal{A}_e, \mathcal{C}_e, \mathcal{R}_e, \mathcal{V}_e)$, where the components are defined as

$$\mathcal{S}_e = \frac{q_1 q_2}{\beta_c q_2\,(1-\rho)+\rho\,(qr_1\beta_c+q_3\beta_a)}, \quad \mathcal{V}_e = \frac{1}{q_4}\left\{(1-\xi)\Gamma+\vartheta\mathcal{S}_e\right\},$$

$$\mathcal{A}_e = \frac{q_1 q_2 q_3\left\{\mathcal{R}_0-1\right\}+\zeta\mathcal{V}_e\rho\beta_c qr_1+\zeta\mathcal{V}_e\beta_c q_2(1-\rho)+\zeta\mathcal{V}_e\rho q_3\beta_a}{\left\{\beta_c q_2(1-\rho)+qr_1\rho\beta_c+q_3\rho\beta_a\right\}\left\{\rho\Gamma\xi\alpha r_1 q+q_2 q_3+\Gamma\xi\alpha q_2(1-\rho)\right\}}, \tag{7}$$

$$\mathcal{C}_e = \frac{\mathcal{A}_e\left\{\beta_a\mathcal{S}_e(1-\rho)+qr_1\right\}}{\beta_c\mathcal{S}_e+\rho\beta_c\mathcal{S}_e+q_3}, \quad \mathcal{R}_e = \frac{1}{\eta}\left\{(1-q)r_1\mathcal{A}_e+r_2\mathcal{C}_e\right\}.$$

It is clear from the second equation of the above system (7) that the endemic state of the proposed epidemic problem continuously depends on the basic reproductive number $R_0$, therefore, it is concluded that the existence of the endemic state is subjected to the condition of $\mathcal{R}_0 > 1$. In the case of $R_0 < 1$, the proposed epidemic problem has only a disease-free state while the endemic state does not exist.

**Lemma 1** *The existence of the endemic state for the model (4) is subject to condition $\mathcal{R}_0 > 1$, otherwise it does not exist.*

The temporal dynamics of the proposed epidemic problem (4)-(5) around the steady-states are described by the following results.

**Theorem 2** *If the value of the threshold parameter is less than unity, i.e., $\mathcal{R}_0 < 1$, then the dynamical system (4) is asymptotically stable around disease-free equilibrium. But if it is greater than unity then the system is stable at the endemic equilibrium.*

Table 1: Parameters and its associated sensitivity indices along with the relative percentage impact on the threshold quantity ($\mathcal{R}_0$)

| Parameters | Indices | % Increase or Decrease | Impact on $\mathcal{R}_0$ |
|---|---|---|---|
| $\beta_a$ | 0.2896 | 10 | 2.8960 % |
| $\beta_c$ | 0.7103 | 10 | 7.1030 % |
| $r_1$ | -0.2321 | 10 | 2.3210 % |
| $r_2$ | -0.0587 | 10 | 0.5870 % |
| $\vartheta$ | -0.9803 | 10 | 9.8030 % |
| $\alpha$ | 0.0014 | 10 | 0.0140 % |

## 4.2 SENSITIVITY ANALYSIS

To analyze the sensitivity of the model parameters and their relative impact on the threshold parameter, $\mathcal{R}_0$, we calculate the sensitivity indices of the model parameters. These indices do not allow us to find the relative impact on basic reproductive numbers only but also will quantify the most sensitive parameters to the disease spreading and control, which is very useful for formulating a control mechanism. To calculate the indices, let us assume the parametric values are as: $\Gamma = 0.343$, $\beta_a = 0.44$, $\beta_c = 0.4570$, $r_2 = 0.0081$, $r_1 = 0.0590$, $\rho = 0.2600$, $q = 0.59$, $\eta_1 = 0.08$, $\eta = 0.01$, $\alpha = 0.02$, $\xi = 0.03$, $\theta = 0.5$. The various epidemic parameters sensitivity indices are calculated and given in Table 1. The sensitivity indices having positive signs show a direct relation with the threshold parameter, $\mathcal{R}_0$, and whenever their values increase or decrease, the value of the threshold parameter will also increase or decrease. on the other hand, the parameters with negative sensitivity indices are inversely proportional to the threshold parameter, $\mathcal{R}_0$. That is, if their values increase, then the value of the threshold parameter will decrease, while if the value of the epidemic parameters decrease then the value of the threshold parameter will increase. It is clear, that the parameters which have a direct relation are $\beta_a$, $\beta_c$, and $\alpha$ with accumulative sensitive index 1.0013. This implies that increase or decrease in the value of these parameters by 10% will increase or decrease the value of the threshold parameter by 10.013%. Similarly, the parameter with negative sensitive indices are $r_1$, $r_2$, and $\vartheta$ having $-0.2321$, $-0.0587$ and $-0.9803$ sensitive indices, respectively. So, increase in the value of $r_1$, $r_2$ and $\vartheta$ by 10% would decrease the value of the basic reproductive number by 2.321%, 0.587% and 9.803%, respectively. We observe that $\beta_c$ and $\vartheta$ are the most sensitive parameters having highest sensitivity indices and significantly affecting the threshold parameter. Keeping in view the sensitivity analysis and temporal dynamic of the model, it is easy to suggest control measure and mechanism for the disease HBV.

## 5 OPTIMAL CONTROL STRATEGIES

Various mathematical techniques are used to characterize optimal control analysis for infectious diseases (Rohani et al., 2008). Optimal control is one of the useful mathematical tool optimizing time-varying control of dynamical systems which has been widely used (Lenhart & Workman, 2007). By analyzing a set of equations illustrating the dynamics of a disease, optimal control theory can mathematically characterize the optimal strategy for a given control method and present insight into the underlying dynamics, without the repeated simulation required to optimize simulation models. We formulate a control mechanism programme for the eradication of HBV transmission. Based on the dynamics and sensitivity, it is quantified that $\beta_c$ and $\vartheta$ are the most sensitive epidemic parameters that significantly affect the threshold parameter as well as the disease propagation. To formulate the control problem, we introduce the following two control measures:

- $\mu_1(t)$ is a time dependent control measure physically describing the treatment of hepatitis B infected population.
- $\mu_2(t)$ is the time dependent control measure representing the vaccination of susceptible population.

The clear goal of the control mechanism is to maximize the number of recovered and vaccinated individuals and meanwhile to minimize the acute and chronic individuals using the above two time

dependent controls, $\mu_1(t)$ and $\mu_2(t)$. Thus, the proposed control problem is the modification of system (4), which takes the following form

$$J = \min \int_0^T \left\{ h_1 \mathcal{A}(t) + h_2 \mathcal{C}(t) + \frac{1}{2} \left( k_1 u_1^2(t) + k_2 u_2^2(t) \right) \right\} dt, \tag{8}$$

subject to

$$\frac{d\mathcal{S}(t)}{dt} = \Gamma\xi \left( 1 - \alpha\mathcal{C}(t) \right) + \zeta\mathcal{V}(t) - \left( \beta_a\mathcal{A}(t) + \beta_c\mathcal{C}(t) + \eta + u_2(t) \right)\mathcal{S}(t),$$

$$\frac{d\mathcal{A}(t)}{dt} = \left( \beta_a\mathcal{A}(t) + \beta_c\mathcal{C}(t) \right)\mathcal{S}(t)\rho - \left( \eta + r_1 + u_1(t) \right)\mathcal{A}(t),$$

$$\frac{d\mathcal{C}(t)}{dt} = (1 - \rho)\left( \beta_a\mathcal{A}(t) + \beta_c\mathcal{C}(t) \right)\mathcal{S}(t) + qr_1\mathcal{A}(t) - \left( u_1(t) + \eta + \eta_1 + r_2 - \Gamma\xi\alpha \right)\mathcal{C}(t), \quad (9)$$

$$\frac{d\mathcal{R}(t)}{dt} = (1 - q)r_1\mathcal{A}(t) + r_2\mathcal{C}(t) + \left( \mathcal{A}(t) + \mathcal{C}(t) \right)u_1(t) - \eta\mathcal{R}(t),$$

$$\frac{d\mathcal{V}(t)}{dt} = (1 - \xi)\Gamma + u_2(t)\mathcal{S}(t) - (\eta + \zeta)\mathcal{V}(t),$$

with initial condition

$$\mathcal{S}(0) > 0, \quad \mathcal{A}(0) \geq 0, \quad \mathcal{C}(0) \geq 0, \quad \mathcal{R}(0) \geq 0, \quad \mathcal{V}(0) > 0, \tag{10}$$

where $h_i$ and $k_i$, $i = 1, 2$ are the weight constants of infected hepatitis B population and measuring the associated cost with treatment and vaccination control measure respectively. The goal of the objective functional (8) is to minimize the population, $\mathcal{A}(t)$ and $\mathcal{C}(t)$ by taking the cost of control parameters. Here, we need to find the optimal measures denoted by $(\mu_1^*, \mu_2^*)$, such that

$$J(\mu_1^*, \mu_2^*) = \min \left\{ J(\mu_1, \mu_2) | \mu_i \in \mathcal{U}, i = 1, 2 \right\}, \tag{11}$$

subject to the model (9). Moreover, assuming that $\mathcal{U}$ is the control set defined as

$$\mathcal{U} := \left\{ (\mu_1, \mu_2) | \mu_i(t) \quad \textit{is Lebesgue measurable on } [0, 1], \quad 0 \leq \mu_i(t) \leq 1, \quad i = 1, 2 \right\}. \tag{12}$$

For these control measures, first we prove their existence. So the control system (9) can be re-written as

$$\frac{dw}{dt} = Kw + M(w), \tag{13}$$

where

$$K = \begin{bmatrix} -(\eta + u_2(t)) & 0 & 0 & -\Gamma\xi\alpha & \zeta \\ 0 & -(\eta + u_1(t)) & 0 & 0 & 0 \\ 0 & qr_1 & -(u_1(t) + \eta + \eta_1 + r_2 - \Gamma\xi\alpha) & 0 & 0 \\ u_2(t) & 0 & 0 & 0 & -\eta \end{bmatrix},$$

$$w = \begin{bmatrix} \mathcal{S}(t) \\ \mathcal{A}(t) \\ \mathcal{C}(t) \\ \mathcal{R}(t) \\ \mathcal{V}(t) \end{bmatrix}, \quad M(w) = \begin{bmatrix} \Gamma\xi - (\beta_a\mathcal{A}(t) + \beta_c\mathcal{C}(t))\mathcal{S}(t) \\ \rho(\beta_a\mathcal{A}(t) + \beta_c\mathcal{C}(t))\mathcal{S}(t) \\ (1 - \rho)(\beta_a\mathcal{A}(t) + \beta_c\mathcal{C}(t))\mathcal{S}(t) \\ 0 \\ (1 - \xi)\Gamma \end{bmatrix}.$$

Setting

$$F(w) = Kw + M(w). \tag{14}$$

The second term on the right hand side of Eq.(14) satisfies

$$| M(w_1) - M(w_2) | \leq m_1 | \mathcal{S}_1(t) - \mathcal{S}_2(t) | + m_2 | \mathcal{A}_2(t) - \mathcal{A}_1(t) | + m_3 | \mathcal{C}_1(t) - \mathcal{C}_2(t) |$$
$$+ m_4 | \mathcal{R}_2(t) - \mathcal{R}_1(t) | + m_5 | \mathcal{V}_1(t) - \mathcal{V}_2(t) |,$$
$$\leq m \{ | \mathcal{S}_1(t) - \mathcal{S}_2(t) | + | \mathcal{A}_1(t) - \mathcal{A}_2(t) | + | \mathcal{C}_1(t) - \mathcal{C}_2(t) |$$
$$+ | \mathcal{R}_2(t) - \mathcal{R}_1(t) | + | \mathcal{V}_1(t) - \mathcal{V}_2(t) | \},$$

where $m = \max\{m_1, m_2, m_3, m_4, m_5\}$ is a positive constant and does not depend on the state variable of the system (9). So, it can be re-written as

$$| F(w_1) - F(w_2) | \le Q | w_1 - w_2 |, \tag{15}$$

where $Q = \max\{m, \|K\|\}$. So, it implies that $F$ is uniformly Lipschitz continuous and by definition of controls $(\mu_1, \mu_2)$ and the state variables $\mathcal{S}(t), \mathcal{V}(t) > 0$, $\mathcal{A}(t), \mathcal{C}(t), \mathcal{R}(t) \ge 0$, imply that the solution of Eq.(13) exists (Birkhoff & Rota, 1978), as stated below.

**Theorem 3** *There exists an optimal solution $\mu^* = (\mu_1^*, \mu_2^*)$ to the problem (8)-(12), contained in $\mathcal{U}$.*

We characterize the optimal solution to the control problem (8)–(12). To this end, we define the associated Lagrangian and Hamiltonian. Let us assume that $s$ is the state variable, $s = (\mathcal{S}, \mathcal{A}, \mathcal{C}, \mathcal{R}, \mathcal{V})$ and $\mu = (\mu_1, \mu_2)$ is the control variable. Then the Lagrangian and the Hamiltonian are respectively described by the following assertions

$$L(s, \mu) = h_1 \mathcal{A} + h_2 \mathcal{C} + \frac{1}{2} \left( k_1 \mu_1^2(t) + k_2 \mu_2^2(t) \right), \quad H(s, \mu, \lambda) = L(s, \mu) + \lambda g(s, \mu), \tag{16}$$

where

$$\lambda = (\lambda_1, \lambda_2, \lambda_3, \lambda_4, \lambda_5), \quad g(s, \mu) = (g_1(s, \mu), \dots g_5(s, \mu)), \tag{17}$$

and

$$g_1(s, \mu) = \Gamma \xi \left( 1 - \alpha \mathcal{C}(t) \right) + \zeta \mathcal{V}(t) - \left( \beta_a \mathcal{A}(t) + \beta_c \mathcal{C}(t) + \eta + \mu_2(t) \right) \mathcal{S}(t),$$
$$g_2(s, \mu) = \left( \beta_a \mathcal{A}(t) \mathcal{S}(t) + \beta_c \mathcal{C}(t) \mathcal{S}(t) \right) \rho - \left( \eta + r_1 + \mu_1(t) \right) \mathcal{A}(t),$$
$$g_3(s, \mu) = (1 - \rho) \left( \beta_a \mathcal{A}(t) + \beta_c \mathcal{C}(t) \right) \mathcal{S}(t) + q r_1 \mathcal{A}(t) - \left( \mu_1(t) + \eta + \eta_1 + r_2 - \Gamma \xi \alpha \right) \mathcal{C}(t),$$
$$g_4(s, \mu) = (1 - q) r_1 \mathcal{A}(t) + r_2 \mathcal{C}(t) + \left( \mathcal{A}(t) + \mathcal{C}(t) \right) \mu_1(t) - \eta \mathcal{R}(t),$$
$$g_5(x, u) = (1 - \xi) \Gamma + \mu_2(t) \mathcal{S}(t) - (\eta + \zeta) \mathcal{V}(t).$$

## 6 NUMERICAL COMPUTATION

This section is devoted to numerical investigation of the model to verify the theoretical results with the help of some graphical visualization using Euler and Runge-Kutta methods. To verify the stability results of the model we use Euler method, while for the optimal control analysis the Runge-Kutta method is used. We will present the model discretization and then discuss the obtained results.

### 6.1 NUMERICAL SIMULATION

The numerical simulations are performed to verify our analytical findings with the help of some computational analysis. To do so, we use the Euler scheme and the fourth-order Runge-Kutta scheme given (due to space limit) in Algorithms 1 and 2 of the appendix, respectively. More precisely, the Euler scheme is used to simulate the proposed model (4) and analyze the stability results. For this, we assume the biological feasible parameters and different non-negative initial population sizes for every class of individuals, as well as the time interval of 0 to 100 units. In case of disease-free state the value of biological parameters are assumed as: $\Gamma = 0.943$, $\xi = 0.05$, $\beta_a = 0.0044$, $\beta_c = 0.00035$, $r_2 = 0.081$, $r_1 = 0.0590$, $\rho = 0.02$, $q = 0.059$, $\eta_1 = 0.012$, $\eta = 0.09$, $\alpha = 0.001$, $\zeta = 0.4$, and $\vartheta = 0.5$. We respectively calculate values of the disease-free state, $\mathcal{D}_f$ and the threshold quantity $(\mathcal{R}_0)$ as $\mathcal{D}_f = (4.011, 0, 0, 0, 2.79)$ and $\mathcal{R}_0 = 0.00157$. We execute the proposed model using the Euler algorithm along with the above parametric values and obtain the simulations carried out as reported in Figure 2. This clearly verifies the analytical findings as stated in Theorem 2. We use the approach of linear stability analysis, that is, perturbing the initial sizes of the compartmental population from the disease-free state while the solution trajectories go to the disease-free equilibrium irrespective of its initial values. This interpretation states that whenever $\mathcal{R}_0 < 1$, each solution curve of the susceptible population takes 60 units of time to reach its equilibrium position of 4.011 as shown in Figure 2a. Similarly, the dynamics of infected (acute & chronic) and the recovered population reveal that the solution trajectories reach zero (vanishes) by taking 35, 30, and 70 units of time as shown in Figures 2b, 2c and 2d respectively. The dynamics of the vaccinated individual are depicted in Figure 2e, which describes that the solution trajectories of the vaccinated population will be non-zero and lead to its equilibrium position by taking 70 units of time, as shown

in Figure 2e. This ensures the stability of the proposed epidemic problem around a disease-free state, while the biological interpretation shows that there will be always susceptible and vaccinated individuals and the infected population vanishes if $\mathcal{R}_0 < 1$. It could be noted that the elimination of hepatitis B is subjected to the condition of $\mathcal{R}_0 < 1$, and therefore it is very necessary to optimize and keep the value of the threshold parameter as low as possible.

However, if this is not the case, that is, $\mathcal{R}_0 > 1$, we assume another set of parameters values as: $\Gamma = 0.943$, $\xi = 0.05$, $\beta_a = 0.0044$, $\beta_c = 0.00035$, $r_2 = 0.081$, $r_1 = 0.0590$, $\rho = 0.02$, $q = 0.059$, $\eta_1 = 0.012$, $\eta = 0.09$, $\alpha = 0.001$, $\zeta = 0.4$, and $\vartheta = 0.5$ to perform stability analysis of the endemic state. We calculate the value of the endemic state and the threshold parameter: $\mathcal{D}_e = (0.531, 3.967, 1.652, 1.725, 1.513)$ and $\mathcal{R}_0 = 1.1364$. Similar to the previous case, the same initial population sizes are taken and the model simulations are performed. We obtain the results as given in Figure 3. This describes the time dynamics of the proposed model at the endemic state. It is very much clear that the susceptible population decreases sharply from the very beginning of infection and leads to its associated endemic position of 0.351 as shown in Figure 3a, while the dynamics of acute, chronic, and recovered population reveals that they are increasing at the initial time of the infection but decreasing then and attains its endemic positions, 3.967, 1.652, and 1.725 as shown in the Figure 3b, Figure 3c, and Figure 3d, respectively. Similarly, the dynamics of vaccinated individuals are represented by Figure 3e which shows that it will attain the endemic position of 1.513 after a unit of time. It is clear from this analysis that the infected (acute & chronic) population persist in the community whenever no proper control measures are implemented. Furthermore, the optimal control problem (8-9) is simulated using Algorithm 2 by considering the set of parameter values corresponds to the endemic state of the proposed model. Moreover, the value of the weight constants are assumed as: $h_1 = 0.6$, $h_2 = 0.9$, $h_3 = 0.44$ and $h_4 = 0.2$, while the time interval is taken to be 0 to 20 units. Here, to see the effect of the optimal measures we plotted the compartmental population of the model with and without control as shown in Figure 4. It is very easy to observe that the goal of applying the control measures: to minimize the infected population and maximize the recovered, and vaccinated population. More precisely, the dynamics of infected population with and without optimal control are visualized in Figure 4b-4c. We note that the number of acutely infected individuals with optimal control are always decreasing from the beginning and vanishes in the first few units of time, while without control it is increasing and always exists. Similarly, the number of chronically infected population are decreasing with optimal control application and reaches to zero, however, increasing without control at the beginning and then decreasing with permanent existence as shown in Figure 4c. The number of recovered and vaccinated population is increasing and then reaches to a certain level of amount when applying the optimal control measures, but decreasing in case of no control as shown in Figure 4d and 4d. We clearly observe the difference between two cases (with optimal control and without optimal control) and believe that the proposed control strategy will be helpful for the elimination of hepatitis B virus elimination.

## 7 CONCLUSION

The formulation of a mathematical model representing the complex biological or physical situation involves some amount of simplifications because of the purpose to describe and predict the essential pattern of the process. The current work demonstrate the detailed derivation and analysis of a model for hepatitis B virus transmission with optimal control. We formulated the model and used a variety of mathematical methods including linear stability approach, basic reproductive number, sensitivity analysis, optimal control theory, Hamiltonian and Lagrangian, Pontryagins principles, numerical simulations to analyze the epidemic problem. We discussed the existence of model solution and showed that the model is feasible biologically and mathematically. We investigated the conditions for stability and performed the sensitivity analysis of the threshold quantity to figure out the most sensitive epidemic parameters. On the basis of sensitivity and stability conditions an optimal control mechanism has been developed in the form of two time dependent control measures, i.e., treatment and vaccination, for the minimization of acute and chronic individuals while maximizing the recovered and vaccinated population. We then conducted the numerical simulations to verify the the theoretical results. It could be concluded that two control measures play an important role to eliminate the infection of hepatitis B virus transmission.

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

# A    PROOFS OF PROPOSITIONS AND THEOREMS

## A.1 PROOF OF PROPOSITION 1

*Proof.* Since, $w = (\mathcal{S}, \mathcal{A}, \mathcal{B}, \mathcal{R}, \mathcal{V})^\top$, and let us assume that

$$
\begin{aligned}
y_{11} &= \beta_a \mathcal{A} + \beta_c \mathcal{C} + \eta + \vartheta, \quad y_{21} = \rho\left(\beta_a \mathcal{A} + \beta_c \mathcal{C}\right), \quad y_{22} = \eta + \gamma_1, \\
y_{31} &= (1 - \rho)\left(\beta_a \mathcal{A} + \beta_c \mathcal{C}\right), \quad y_{33} = \eta + \eta_1 + \gamma_2 - \alpha \Gamma \xi, \quad y_{55} = \eta + \zeta,
\end{aligned}
\tag{18}
$$

then the proposed system (4) takes the form

$$
\frac{dw}{dt} = \mathcal{A}w + \mathcal{C},
$$

where

$$
\mathcal{A} = \begin{pmatrix}
-y_{11} & 0 & -\alpha \Gamma \xi & 0 & \zeta \\
y_{21} & -y_{22} & 0 & 0 & 0 \\
y_{31} & qr_1 & -y_{33} & 0 & 0 \\
0 & r_1(1-q) & r_2 & -\eta & 0 \\
\vartheta & 0 & 0 & 0 & -y_{55}
\end{pmatrix}, \quad
\mathcal{C} = \begin{pmatrix}
\Gamma \xi \\
0 \\
0 \\
0 \\
\Gamma(1 - \xi)
\end{pmatrix}.
$$

Clearly, $\mathcal{C}$ is non-negative, and $\mathcal{A}$ is a Metzler matrix, which is enough to show that the model (4) is invariant positively in $R_+^5$.

## A.2 PROOF OF PROPOSITION 2

*Proof.* Let $[0, +\infty)$ is the interval of solution for system (4), then the solution $\mathcal{S}(t)$ for the first equation looks like

$$
\begin{aligned}
\mathcal{S}(t) = \exp &\left\{ -(\eta + \vartheta)t - \int_0^t \left(\beta_a \mathcal{A}(x) + \beta_c \mathcal{C}(x)\right) dx \right\} \\
&\times \int_0^t \left\{ \Gamma \zeta \left(1 - \alpha \mathcal{C}(x)\right) + \zeta \mathcal{V}(x) \right\} \times \exp \left\{ (\eta + \vartheta)x + \int_0^x \left(\beta_a(y) + \beta_c(y)\right) dy \right\} dx \\
&+ \mathcal{S}(0) \exp \left\{ -(\eta + \vartheta)t - \int_0^t \left(\beta_a \mathcal{A}(x) + \beta_c \mathcal{C}(x)\right) dx \right\} > 0.
\end{aligned}
$$

Like wise, the solution $\mathcal{A}(t)$ may take the form

$$
\begin{aligned}
\mathcal{A}(t) = \exp &\left\{ -(\eta + r_1)t + \int_0^t \rho \beta_a \mathcal{S}(x) dx \right\} \int_0^t \rho \beta_c \mathcal{C}(x) \exp \left\{ (\eta + r_1)x - \int_0^x \rho \beta_a(y) \mathcal{S}(y) dy \right\} dx \\
&+ \mathcal{A}(0) \exp \left\{ -(\eta + r_1)t + \int_0^t \rho \beta_a \mathcal{S}(x) dx \right\} \geq 0,
\end{aligned}
$$

which describe that $S(t) > 0$ and $A(t) \geq 0 \,\forall\, t \in [0, +\infty)$. In a similar way it is easy to perform that $\mathcal{C} \geq 0$, $\mathcal{R} \geq 0$ and $\mathcal{V} > 0$.

A.3 PROOF OF THEOREM 2

*Proof.* Obviously, from the fourth equation of the model (4), it is clear that $\mathcal{R}(t)$ appears only in the fourth equation, so we can discuss the dynamics of the reduced system. Thus, using the linear stability analysis (2), the variational matrix of system (4) around disease-free state becomes

$$J|_{D_f} = \begin{pmatrix} -q_1 & -\beta_a \mathcal{S}_f & -\beta_c \mathcal{S}_f - \Gamma \xi \alpha & \zeta \\ 0 & \rho \beta_a \mathcal{S}_f - q_2 & \rho \beta_c \mathcal{S}_f & 0 \\ 0 & q r_1 & (1-\rho)\beta_c \mathcal{S}_f - q_3 & 0 \\ \vartheta & 0 & 0 & q_4 \end{pmatrix}. \tag{19}$$

Let $\mathcal{P}(x)$ is the characteristic polynomial of the matrix, $J|_{D_f}$, then

$$\mathcal{P}(x) = a_0 x^4 + a_1 x^3 + a_2 x^2 + a_3 x + a_4, \tag{20}$$

where

$$\begin{aligned}
a_0 &= 1, \quad a_1 = q_1 + q_4 + q_3 \left(1 - \mathcal{R}_1\right) + q_2 \left(1 - \mathcal{R}_2\right), \\
a_2 &= \eta \left(\eta + \zeta + \vartheta\right) + q_3 \left(q_1 + q_4\right) \left(1 - \mathcal{R}_1\right) + q_2 \left(q_1 + q_4\right) \left(1 - \mathcal{R}_2\right) + q_2 q_3 \left(1 - \mathcal{R}_0\right) \\
&\quad + \mathcal{S}_f^2 \beta_a \beta_c \rho \left(1 - \rho\right), \\
a_3 &= q_1 q_2 q_3 \left(1 - \mathcal{R}_0\right) + q_3 \eta \left(\eta + \zeta + \vartheta\right) \left(1 - \mathcal{R}_1\right) + q_2 \eta \left(\eta + \zeta + \vartheta\right) \left(1 - \mathcal{R}_0\right) \\
&\quad + \mathcal{S}_f^2 \beta_a \beta_c \rho \left(1 - \rho\right) - \mathcal{S}_f^2 \left\{\beta_c q_2 q_4 \left(1 - \rho\right) + \beta_a q_3 q_4 \rho + \beta_c q q_1 r_1 \rho + \beta_c q q_4 r_1 \rho\right\}, \\
a_4 &= q_1 q_2 q_3 q_4 \left(1 - \mathcal{R}_0\right) + \mathcal{S}_f \beta_c q_2 \vartheta \zeta \left(1 - \rho\right) + \mathcal{S}_f^2 \beta_a \beta_c q_1 q_4 \rho \left(1 - \rho\right) + \mathcal{S}_f^2 \beta_a \beta_c \rho^2 \vartheta \zeta \\
&\quad + \mathcal{S}_f \beta_a q_3 \rho \vartheta \zeta + \mathcal{S}_f \beta_c q r_1 \rho \vartheta \zeta - q_2 q_3 \vartheta \zeta - \mathcal{S}_f^2 \beta_a \beta_c \rho \vartheta \zeta.
\end{aligned}$$

The co-efficient $a_i$'s reveals that, whenever, $\mathcal{R}_0 < 1$ and $a_3$, and $a_4$ are positive, then the Routh-Hurwitz criterion implies that the roots of the Eq.(20) are negative or having negative real parts, and so the model is locally asymptotically stable under the condition of $\mathcal{R}_0 < 1$ and $a_3$, $a_4 > 0$.

In a similar fashion, it can be shown that the model is stable at the endemic equilibrium if the value of the threshold quantity is greater than unity.

A.4 PROOF OF THEOREM 3

*Proof.* Since, the state and control functions are non-negative while the control set (12) is closed and convex. The system (9) is bounded, and ensures the compactness. Also the integrand in the Eq.(8) is convex with respect to the control measure $\mu_1(t)$ and $\mu_2(t)$. Thus, it proves the conclusion of optimal controls $(\mu_1^*, \mu_2^*)$ existence.

A.5 PROOF OF THEOREM 4

We use Pontryagin's Maximum Principle to obtain an optimal solution and thus we have the assertion given below.

**Theorem 4** *Let $\mathcal{S}^*$, $\mathcal{A}^*$, $\mathcal{C}^*$, $\mathcal{R}^*$ and $\mathcal{V}^*$ be optimal states associated with the optimal controls $(u_1^*, u_2^*)$ for (8)-(12), then $\lambda_i(t)$ (adjoint variables), $i = 1, \ldots, 5$ exist, and satisfy*

$$\lambda_1'(t) = \left\{\lambda_1(t) - \rho \lambda_2(t) - (1-\rho)\lambda_3(t)\right\} \left\{\beta_a \mathcal{A}^*(t) + \beta_c \mathcal{C}^*(t)\right\} + \lambda_1(t) \left\{\eta + u_2^*(t)\right\} - u_2^*(t)\lambda_5(t),$$

$$\begin{aligned}
\lambda_2'(t) = -h_1 + \left\{\lambda_1(t) - \rho \lambda_2(t) - (1-\rho)\lambda_3(t)\right\} \beta_a \mathcal{S}^*(t) + \lambda_2(t) \left\{\eta + r_1 + u_1^*(t)\right\} \\
- \lambda_3(t) q r_1 - \lambda_4(t)(1-q)r_1 - \lambda_4(t)u_1^*(t),
\end{aligned}$$

$$\begin{aligned}
\lambda_3'(t) = -h_2 + \lambda_1(t)\Gamma \xi \alpha + \left\{\lambda_1(t) - \rho \lambda_2(t) - (1-\rho)\lambda_3(t)\right\} \beta_c \mathcal{S}^*(t) \\
+ \lambda_3(t) \left\{u_1^*(t) + \eta + \eta_1 + r_2 - \Gamma \xi \alpha\right\} + \lambda_4(t) \left\{r_2 + u_1^*(t)\right\},
\end{aligned}$$

$$\lambda_4'(t) = \eta \lambda_4(t), \quad \lambda_5'(t) = \left\{\lambda_5(t) - \lambda_1(t)\right\} \zeta + \eta \lambda_5(t),$$

*with terminal (transversality) conditions*

$$\lambda_i(T) = 0.$$

*Moreover, the optimal value of controls are*

$$\mu_1^*(t) = \max\left\{\min\left\{\frac{1}{k_1}(\lambda_2(t) - \lambda_4(t))\mathcal{A}^*(t) + (\lambda_3(t) - \lambda_4(t))\mathcal{C}^*(t), 0\right\}, 1\right\},$$

$$\mu_2^*(t) = \max\left\{\min\left\{\frac{1}{k_2}(\lambda_1(t) - \lambda_5(t))\mathcal{S}^*(t), 0\right\}, 1\right\}.$$

*Proof.* The adjoint system $\left(\lambda_1^{'}(t), \lambda_2^{'}(t), \lambda_3^{'}(t), \lambda_4^{'}(t), \lambda_5^{'}(t)\right)$ has been derived by the direct use of the Pontryagin Principle, while terminal (transversal) conditions are obtained from the use of the transversal condition. Moreover to derive the optimal values, $u_1^*$ and $u_2^*$, the Hamiltonian has been partially differentiated with respect to the control functions $u_1$ and $u_2$ respectively, and then equating $\frac{\partial H}{\partial u_i}$ to zero, and solving. We then use the maximality conditions (3), and obtain the optimal value of the control variables.

## B  DISCRETIZATION OF THE MODEL AND ALGORITHMS FOR NUMERICAL SIMULATIONS

We shall present the temporal dynamics of the proposed model by creating algorithms using the Euler and Runge-Kutta methods. Notice that the proposed epidemic problem contain five population groups and we will apply the algorithm of Euler and Runge-Kutta method to every group of population in order to present the dynamics of every compartment. For the shake of simplicity, let us assume some notations

$$\Phi_1 = \Gamma\xi\left(1 - \alpha\mathcal{C}_i\right) + \zeta\mathcal{V}_i - \left(\beta_a\mathcal{A}_i + \beta_c\mathcal{C}_i + \vartheta + \eta\right)\mathcal{S}_i,$$
$$\Phi_2 = \left(\beta_a\mathcal{A}_i\mathcal{S}_i + \beta_c\mathcal{C}_i\mathcal{S}_i\right)\rho - \left(r_1 + \eta\right)\mathcal{A}_i,$$
$$\Phi_3 = \left(\beta_a\mathcal{A}_i + \beta_c\mathcal{C}_i\right)\mathcal{S}_i\left(1 - \rho\right) + qr_1\mathcal{A}_i - \left(\eta + \eta_1 + r_2 - \Gamma\xi\alpha\right)\mathcal{C}_i,$$
$$\Phi_5 = (1 - q)r_1\mathcal{A}_i + r_2\mathcal{C}_i - \eta\mathcal{R}_i, \quad \Phi_4 = (1 - \xi)\Gamma + \vartheta\mathcal{S}_i - (\eta + \zeta)\mathcal{V}_i,$$

then the algorithm for the model (4) solution with the aid of Euler method as concluded in the Algorithm 1. To present the algorithms for the optimal control problem, we use the forward Runge-

---

**Algorithm 1** Euler Method (EM)

---

1: **Input:** Endpoints $t_0$, $t_{\max}$, integer $n$, parametric values, initial conditions
2: **Output:** approximation $\mathcal{S}$, $\mathcal{A}$, $\mathcal{C}$, $\mathcal{R}$, $\mathcal{V}$ at $(n + 1)$ values of $t$
3: **Parameters and Initial Conditions:** Setting the values for $\Gamma$, $\xi$, $\beta_a$, $\beta_c$, $r_1$, $r_2$, $\rho$, $q$, $\eta$, $\eta_1$ $\alpha$, $\zeta$, $\vartheta$, and for the initial sizes $\mathcal{S}(0)$, $\mathcal{A}(0)$, $\mathcal{C}(0)$, $\mathcal{R}(0)$, $\mathcal{V}(0)$
4: **for** $i = 1, \cdots, n$ **do**
5:   **Recursive Formula:**

$$t_i = t_0 + ih, \quad \mathcal{S}_{i+1} = \mathcal{S}_i + h\Phi_1\left(\mathcal{S}_i, \mathcal{A}_i, \mathcal{C}_i, \mathcal{V}_i\right), \quad \mathcal{A}_{i+1} = \mathcal{A}_i + h\Phi_2\left(\mathcal{S}_i, \mathcal{A}_i, \mathcal{C}_i\right),$$
$$\mathcal{C}_{i+1} = \mathcal{C}_i + h\Phi_3\left(\mathcal{S}_i, \mathcal{A}_i, \mathcal{C}_i\right), \quad \mathcal{R}_{i+1} = \mathcal{R}_i + h\Phi_4\left(\mathcal{A}_i, \mathcal{C}_i\right), \quad \mathcal{V}_{i+1} = \mathcal{V}_i + h\Phi_5\left(\mathcal{S}_i, \mathcal{V}_i\right),$$

6: **end for**
7: **Output** $(t_i, \mathcal{S}_{i+1}, \mathcal{A}_{i+1}, \mathcal{C}_{i+1}, \mathcal{R}_{i+1}, \mathcal{V}_{i+1})$

---

Kutta method and assume the following notations:

$$\Phi_1^{\mathcal{S}} = \Gamma\xi\left\{1 - \alpha\mathcal{C}_i\right\} + \zeta\mathcal{V}_i - \left\{\beta_a\mathcal{A}_i + \beta_c\mathcal{C}_i + \vartheta + \eta\right\}\mathcal{S}_i,$$

$$\Phi_1^{\mathcal{A}} = \left\{\beta_a\mathcal{A}_i\mathcal{S}_i + \beta_c\mathcal{C}_i\mathcal{S}_i\right\}\rho - \left\{r_1 + \eta\right\}\mathcal{A}_i,$$

$$\Phi_1^{\mathcal{C}} = \left\{\beta_a\mathcal{A}_i + \beta_c\mathcal{C}_i\right\}\mathcal{S}_i\left(1 - \rho\right) + qr_1\mathcal{A}_i - \left\{\eta + \eta_1 + r_2 - \Gamma\xi\alpha\right\}\mathcal{C}_i,$$

$$\Phi_1^{\mathcal{R}} = (1 - q)r_1\mathcal{A}_i + r_2\mathcal{C}_i - \eta\mathcal{R}_i,$$

$$\Phi_1^{\mathcal{V}} = (1 - \xi)\,\Gamma + \vartheta\mathcal{S}_i - \left\{\eta + \zeta\right\}\mathcal{V}_i,$$

$$\Phi_2^{\mathcal{S}} = \Gamma\xi\left\{1 - \alpha\left(\mathcal{C}_i + \frac{h\Phi_1^{\mathcal{C}}}{2}\right)\right\} - \left\{\beta_a\left(\mathcal{A}_i + \frac{h\Phi_1^{\mathcal{A}}}{2}\right) + \beta_c\left(\mathcal{C}_i + \frac{h\Phi_1^{\mathcal{C}}}{2}\right) + \vartheta + \eta\right\}$$
$$\left\{\mathcal{S}_i + \frac{h\Phi_1^{\mathcal{S}}}{2}\right\} + \zeta\left\{\mathcal{V}_i + \frac{h\Phi_1^{\mathcal{V}}}{2}\right\},$$

$$\Phi_2^{\mathcal{A}} = \left\{\beta_a\left(\mathcal{A}_i + \frac{h\Phi_1^{\mathcal{A}}}{2}\right) + \beta_c\left(\mathcal{C}_i + \frac{h\Phi_1^{\mathcal{C}}}{2}\right)\right\}\left\{\mathcal{S}_i + \frac{h\Phi_1^{\mathcal{S}}}{2}\right\}\rho - (r_1 + \eta)\left\{\mathcal{A}_i + \frac{h\Phi_1^{\mathcal{A}}}{2}\right\},$$

$$\Phi_2^{\mathcal{C}} = \left\{\beta_a\left(\mathcal{A}_i + \frac{h\Phi_1^{\mathcal{A}}}{2}\right) + \beta_c\left(\mathcal{C}_i + \frac{h\Phi_1^{\mathcal{C}}}{2}\right)\right\}\left\{\mathcal{S}_i + \frac{h\Phi_1^{\mathcal{S}}}{2}\right\}(1 - \rho) + qr_1\left\{\mathcal{A}_i + \frac{h\Phi_1^{\mathcal{A}}}{2}\right\}$$
$$- \left\{\eta + \eta_1 + r_2 - \Gamma\xi\alpha\right\}\left\{\mathcal{C}_i + \frac{h\Phi_1^{\mathcal{C}}}{2}\right\},$$

$$\Phi_2^{\mathcal{R}} = (1 - q)r_1\left\{\mathcal{A}_i + \frac{h\Phi_1^{\mathcal{A}}}{2}\right\} + r_2\left\{\mathcal{C}_i + \frac{h\Phi_1^{\mathcal{C}}}{2}\right\} - \eta\left\{\mathcal{R}_i + \frac{h\Phi_1^{\mathcal{R}}}{2}\right\},$$

$$\Phi_2^{\mathcal{V}} = (1 - \xi)\,\Gamma + \vartheta\left\{\mathcal{S}_i + \frac{h\Phi_1^{\mathcal{S}}}{2}\right\} - (\eta + \zeta)\left\{\mathcal{V}_i + \frac{h\Phi_1^{\mathcal{V}}}{2}\right\},$$

$$\Phi_3^{\mathcal{S}} = \Gamma\xi\left\{1 - \alpha\left(\mathcal{C}_i + \frac{h\Phi_2^{\mathcal{C}}}{2}\right)\right\} - \left\{\beta_a\left(\mathcal{A}_i + \frac{h\Phi_2^{\mathcal{A}}}{2}\right) + \beta_c\left(\mathcal{C}_i + \frac{h\Phi_2^{\mathcal{C}}}{2}\right) + \vartheta + \eta\right\}$$
$$\left\{\mathcal{S}_i + \frac{h\Phi_2^{\mathcal{S}}}{2}\right\} + \zeta\left\{\mathcal{V}_i + \frac{h\Phi_2^{\mathcal{V}}}{2}\right\},$$

$$\Phi_3^{\mathcal{A}} = \left\{\beta_a\left(\mathcal{A}_i + \frac{h\Phi_2^{\mathcal{A}}}{2}\right) + \beta_c\left(\mathcal{C}_i + \frac{h\Phi_2^{\mathcal{C}}}{2}\right)\right\}\left\{\mathcal{S}_i + \frac{h\Phi_2^{\mathcal{S}}}{2}\right\}\rho - (r_1 + \eta)\left\{\mathcal{A}_i + \frac{h\Phi_2^{\mathcal{A}}}{2}\right\},$$

$$\Phi_3^{\mathcal{C}} = \left\{\beta_a\left(\mathcal{A}_i + \frac{h\Phi_2^{\mathcal{A}}}{2}\right) + \beta_c\left(\mathcal{C}_i + \frac{h\Phi_2^{\mathcal{C}}}{2}\right)\right\}\left\{\mathcal{S}_i + \frac{h\Phi_2^{\mathcal{S}}}{2}\right\}(1 - \rho) + qr_1\left\{\mathcal{A}_i + \frac{h\Phi_2^{\mathcal{A}}}{2}\right\}$$
$$- (\eta + \eta_1 + r_2 - \Gamma\xi\alpha)\left\{\mathcal{C}_i + \frac{h\Phi_2^{\mathcal{C}}}{2}\right\},$$

$$\Phi_3^{\mathcal{R}} = (1 - q)r_1\left\{\mathcal{A}_i + \frac{h\Phi_2^{\mathcal{A}}}{2}\right\} + r_2\left\{\mathcal{C}_i + \frac{h\Phi_2^{\mathcal{C}}}{2}\right\} - \eta\left\{\mathcal{R}_i + \frac{h\Phi_2^{\mathcal{R}}}{2}\right\},$$

$$\Phi_3^{\mathcal{V}} = (1 - \xi)\,\Gamma + \vartheta\left\{\mathcal{S}_i + \frac{h\Phi_2^{\mathcal{S}}}{2}\right\} - (\eta + \zeta)\left\{\mathcal{V}_i + \frac{h\Phi_2^{\mathcal{V}}}{2}\right\},$$

$$\Phi_4^{\mathcal{S}} = \Gamma\xi\left\{1 - \alpha\left(\mathcal{C}_i + \frac{h\Phi_3^{\mathcal{C}}}{2}\right)\right\} - \left\{\beta_a\left(\mathcal{A}_i + \frac{h\Phi_3^{\mathcal{A}}}{2}\right) + \beta_c\left(\mathcal{C}_i + \frac{h\Phi_3^{\mathcal{C}}}{2}\right) + \vartheta + \eta\right\}$$
$$\left\{\mathcal{S}_i + \frac{h\Phi_3^{\mathcal{S}}}{2}\right\} + \zeta\left\{\mathcal{V}_i + \frac{h\Phi_3^{\mathcal{V}}}{2}\right\},$$

$$\Phi_4^{\mathcal{A}} = \left\{\beta_a\left(\mathcal{A}_i + \frac{h\Phi_3^{\mathcal{A}}}{2}\right) + \beta_c\left(\mathcal{C}_i + \frac{h\Phi_3^{\mathcal{C}}}{2}\right)\right\}\left\{\mathcal{S}_i + \frac{h\Phi_3^{\mathcal{S}}}{2}\right\}\rho - (r_1 + \eta)\left\{\mathcal{A}_i + \frac{h\Phi_3^{\mathcal{A}}}{2}\right\},$$

$$\Phi_4^{\mathcal{C}} = \left\{\beta_a\left(\mathcal{A}_i + \frac{h\Phi_3^{\mathcal{A}}}{2}\right) + \beta_c\left(\mathcal{C}_i + \frac{h\Phi_3^{\mathcal{C}}}{2}\right)\right\}\left\{\mathcal{S}_i + \frac{h\Phi_3^{\mathcal{S}}}{2}\right\}(1 - \rho) + qr_1\left\{\mathcal{A}_i + \frac{h\Phi_3^{\mathcal{A}}}{2}\right\}$$
$$- (\eta + \eta_1 + r_2 - \Gamma\xi\alpha)\left\{\mathcal{C}_i + \frac{h\Phi_3^{\mathcal{C}}}{2}\right\},$$

$$\Phi_4^{\mathcal{R}} = (1 - q)r_1\left\{\mathcal{A}_i + \frac{h\Phi_3^{\mathcal{A}}}{2}\right\} + r_2\left\{\mathcal{C}_i + \frac{h\Phi_3^{\mathcal{C}}}{2}\right\} - \eta\left\{\mathcal{R}_i + \frac{h\Phi_3^{\mathcal{R}}}{2}\right\},$$

$$\Phi_4^{\mathcal{V}} = (1 - \xi)\,\Gamma + \vartheta\left\{\mathcal{S}_i + \frac{h\Phi_3^{\mathcal{S}}}{2}\right\} - (\eta + \zeta)\left\{\mathcal{V}_i + \frac{h\Phi_3^{\mathcal{V}}}{2}\right\}.$$

Similarly by backward Runge-Kutta method, we have

$$\Phi_{\lambda_{11}} = \{\lambda_{1j} - \rho\lambda_{2j} - (1-\rho)\lambda_{3j}\}\{\beta_a\mathcal{A}_j^* + \beta_c\mathcal{C}_j^*\} + \lambda_{1j}\{\eta + \mu_{2j}^*\} - \mu_{2j}^*\lambda_{5j},$$

$$\Phi_{\lambda_{21}} = -h_1 + \{\lambda_{1j} - \rho\lambda_{2j} - (1-\rho)\lambda_{3j}\}\beta_a\mathcal{S}_j^* + \lambda_{2j}\{\eta + r_1 + \mu_{1j}^*\}$$
$$- \lambda_{3j}qr_1 - \lambda_{4j}(1-q)r_1 - \lambda_{4j}\mu_{1j}^*,$$

$$\Phi_{\lambda_{31}} = -h_2 + \lambda_{1j}\Gamma\xi\alpha + \{\lambda_{1j} - \rho\lambda_{2j} - (1-\rho)\lambda_{3j}\}\beta_c\mathcal{S}_j^*$$
$$+ \lambda_{3j}\{u_{1j}^* + \eta + \eta_1 + r_2 - \Gamma\xi\alpha\} + \lambda_{4j}\{r_2 + \mu_{1j}^*\},$$

$$\Phi_{\lambda_{41}} = \eta\lambda_{4j}, \quad \Phi_{\lambda_{51}} = \{\lambda_{5j} - \lambda_{1j}\}\zeta + \eta\lambda_{5j},$$

$$\Phi_{\lambda_{12}} = \left[\left\{\lambda_{1j} - \frac{h\Phi_{\lambda_{11}}}{2}\right\} - \rho\left\{\lambda_{2j} - \frac{h\Phi_{\lambda_{21}}}{2}\right\} - (1-\rho)\left\{\lambda_{3j} - \frac{h\Phi_{\lambda_{31}}}{2}\right\}\right]$$
$$\left\{\frac{1}{2}\{\beta_a(\mathcal{A}_j^* + \mathcal{A}_{j-1}^*) + \beta_c(\mathcal{C}_j^* + \mathcal{C}_{j-1}^*)\}\right\} + \left\{\lambda_{1j} - \frac{h\Phi_{\lambda_{11}}}{2}\right\}$$
$$\left\{\eta + \frac{1}{2}(\mu_{2j}^* + \mu_{2(j-1)}^*)\right\} - \frac{1}{2}(\mu_{2j}^* + \mu_{2(j-1)}^*)\left\{\lambda_{5j} - \frac{h\Phi_{\lambda_{51}}}{2}\right\},$$

$$\Phi_{\lambda_{22}} = \left\{\left(\lambda_{1j} - \frac{h\Phi_{\lambda_{11}}}{2}\right) - \rho\left(\lambda_{2j} - \frac{h\Phi_{\lambda_{21}}}{2}\right) - (1-\rho)\left(\lambda_{3j} - \frac{h\Phi_{\lambda_{31}}}{2}\right)\right\}\beta_a\left\{\frac{1}{2}(\mathcal{S}_j^* + \mathcal{S}_{j-1}^*)\right\}$$
$$+ \left\{\lambda_{2j} - \frac{h\Phi_{\lambda_{21}}}{2}\right\}\left\{\eta + r_1 + \frac{1}{2}(u_{1j}^* + u_{1(j-1)}^*)\right\} - \left\{\lambda_{3j} - \frac{h\Phi_{\lambda_{31}}}{2}\right\}qr_1$$
$$- \left\{\lambda_{4j} - \frac{h\Phi_{\lambda_{41}}}{2}\right\}(1-q)r_1 - \left\{\lambda_{4j} - \frac{h\Phi_{\lambda_{41}}}{2}\right\}\left\{\frac{1}{2}(\mu_{1j}^* + \mu_{1(j-1)}^*)\right\} - h_1,$$

$$\Phi_{\lambda_{32}} = -h_2 + \left\{\lambda_{1j} - \frac{h\Phi_{\lambda_{11}}}{2}\right\}\Gamma\xi\alpha + \left\{\left(\lambda_{1j} - \frac{h\Phi_{\lambda_{11}}}{2}\right) - \rho\left(\lambda_{2j} - \frac{h\Phi_{\lambda_{21}}}{2}\right)\right.$$
$$\left. - (1-\rho)\left(\lambda_{3j} - \frac{h\Phi_{\lambda_{31}}}{2}\right)\right\}\left\{\frac{\beta_c}{2}(\mathcal{S}_j^* + \mathcal{S}_{j-1}^*)\right\} + \left\{\lambda_{4j} - \frac{h\Phi_{\lambda_{41}}}{2}\right\}\left\{r_2 + \frac{1}{2}(\mu_{1j}^* + \mu_{1(j-1)}^*)\right\}$$
$$+ \left\{\lambda_{3j} - \frac{h\Phi_{\lambda_{31}}}{2}\right\}\left\{\frac{1}{2}(u_{1j}^* + u_{1(j-1)}^*) + \eta + \eta_1 + r_2 - \Gamma\xi\alpha\right\},$$

$$\Phi_{\lambda_{42}} = \eta\left\{\lambda_{4j} - \frac{h\Phi_{\lambda_{41}}}{2}\right\}, \quad \Phi_{\lambda_{52}} = \left\{\left(\lambda_{5j} - \frac{h\Phi_{\lambda_{51}}}{2}\right) - \left(\lambda_{1j} - \frac{h\Phi_{\lambda_{11}}}{2}\right)\right\}\zeta + \eta\left\{\lambda_{5j} - \frac{h\Phi_{\lambda_{51}}}{2}\right\},$$

$$\Phi_{\lambda_{13}} = \left\{\left(\lambda_{1j} - \frac{h\Phi_{\lambda_{12}}}{2}\right) - \rho\left(\lambda_{2j} - \frac{h\Phi_{\lambda_{22}}}{2}\right) - (1-\rho)\left(\lambda_{3j} - \frac{h\Phi_{\lambda_{32}}}{2}\right)\right\}$$
$$\left\{\frac{1}{2}\{\beta_a(\mathcal{A}_j^* + \mathcal{A}_{j-1}^*) + \beta_c(\mathcal{C}_j^* + \mathcal{C}_{j-1}^*)\}\right\} + \left\{\lambda_{1j} - \frac{h\Phi_{\lambda_{12}}}{2}\right\}\left\{\eta + \frac{1}{2}(\mu_{2j}^* + \mu_{2(j-1)}^*)\right\}$$
$$- \frac{1}{2}(\mu_{2j}^* + \mu_{2(j-1)}^*)\left\{\lambda_{5j} - \frac{h\Phi_{\lambda_{52}}}{2}\right\},$$

$$\Phi_{\lambda_{23}} = \left\{\left(\lambda_{1j} - \frac{h\Phi_{\lambda_{12}}}{2}\right) - \rho\left(\lambda_{2j} - \frac{h\Phi_{\lambda_{22}}}{2}\right) - (1-\rho)\left(\lambda_{3j} - \frac{h\Phi_{\lambda_{32}}}{2}\right)\right\}\beta_a\left\{\frac{1}{2}(\mathcal{S}_j^* + \mathcal{S}_{j-1}^*)\right\}$$
$$+ \left\{\lambda_{2j} - \frac{h\Phi_{\lambda_{22}}}{2}\right\}\left\{\eta + r_1 + \frac{1}{2}(u_{1j}^* + u_{1(j-1)}^*)\right\} - \left\{\lambda_{3j} - \frac{h\Phi_{\lambda_{32}}}{2}\right\}qr_1$$
$$- \left\{\lambda_{4j} - \frac{h\Phi_{\lambda_{42}}}{2}\right\}(1-q)r_1 - \left\{\lambda_{4j} - \frac{h\Phi_{\lambda_{42}}}{2}\right\}\left\{\frac{1}{2}(\mu_{1j}^* + \mu_{1(j-1)}^*)\right\} - h_1,$$

$$\Phi_{\lambda_{33}} = -h_2 + \left\{\lambda_{1j} - \frac{h\Phi_{\lambda_{12}}}{2}\right\}\Gamma\xi\alpha + \left\{\left(\lambda_{1j} - \frac{h\Phi_{\lambda_{12}}}{2}\right) - \rho\left(\lambda_{2j} - \frac{h\Phi_{\lambda_{22}}}{2}\right)\right.$$
$$\left. - (1-\rho)\left(\lambda_{3j} - \frac{h\Phi_{\lambda_{32}}}{2}\right)\right\}\left\{\frac{\beta_c}{2}(\mathcal{S}_j^* + \mathcal{S}_{j-1}^*)\right\} + \left\{\lambda_{4j} - \frac{h\Phi_{\lambda_{42}}}{2}\right\}$$
$$\left\{r_2 + \frac{1}{2}(\mu_{1j}^* + \mu_{1(j-1)}^*)\right\} + \left\{\lambda_{3j} - \frac{h\Phi_{\lambda_{32}}}{2}\right\}\left\{\frac{1}{2}(u_{1j}^* + u_{1(j-1)}^*) + \eta + \eta_1 + r_2 - \Gamma\xi\alpha\right\},$$

$$\Phi_{\lambda_{43}} = \eta\left\{\lambda_{4j} - \frac{h\Phi_{\lambda_{42}}}{2}\right\}, \Phi_{\lambda_{53}} = \left\{\left(\lambda_{5j} - \frac{h\Phi_{\lambda_{52}}}{2}\right) - \left(\lambda_{1j} - \frac{h\Phi_{\lambda_{12}}}{2}\right)\right\}\zeta + \eta\left\{\lambda_{5j} - \frac{h\Phi_{\lambda_{52}}}{2}\right\},$$

$$\Phi_{\lambda_{14}} = \left\{ \left( \lambda_{1j} - \frac{h\Phi_{\lambda_{13}}}{2} \right) - \rho \left( \lambda_{2j} - \frac{h\Phi_{\lambda_{23}}}{2} \right) - (1-\rho) \left( \lambda_{3j} - \frac{h\Phi_{\lambda_{33}}}{2} \right) \right\}$$

$$\left\{ \beta_a \mathcal{A}^*_{j-1} + \beta_c \mathcal{C}^*_{j-1} \right\} + \left\{ \lambda_{1j} - \frac{h\Phi_{\lambda_{13}}}{2} \right\} \left\{ \eta + \mu^*_{2(j-1)} \right\} - \mu^*_{2(j-1)} \left\{ \lambda_{5j} - \frac{h\Phi_{\lambda_{53}}}{2} \right\},$$

$$\Phi_{\lambda_{24}} = \left\{ \left( \lambda_{1j} - \frac{h\Phi_{\lambda_{13}}}{2} \right) - \rho \left( \lambda_{2j} - \frac{h\Phi_{\lambda_{23}}}{2} \right) - (1-\rho) \left( \lambda_{3j} - \frac{h\Phi_{\lambda_{33}}}{2} \right) \right\} \beta_a \mathcal{S}^*_{j-1}$$

$$+ \left\{ \lambda_{2j} - \frac{h\Phi_{\lambda_{23}}}{2} \right\} \left\{ \eta + r_1 + u^*_{1(j-1)} \right\} - \left\{ \lambda_{3j} - \frac{h\Phi_{\lambda_{33}}}{2} \right\} qr_1$$

$$- \left\{ \lambda_{4j} - \frac{h\Phi_{\lambda_{43}}}{2} \right\} (1-q)r_1 - \left\{ \lambda_{4j} - \frac{h\Phi_{\lambda_{43}}}{2} \right\} \mu^*_{1(j-1)} - h_1,$$

$$\Phi_{\lambda_{34}} = -h_2 + \left( \lambda_{1j} - \frac{h\Phi_{\lambda_{13}}}{2} \right) \Gamma\xi\alpha + \left\{ \left( \lambda_{1j} - \frac{h\Phi_{\lambda_{13}}}{2} \right) - \rho \left\{ \lambda_{2j} - \frac{h\Phi_{\lambda_{23}}}{2} \right\} \right.$$

$$\left. - (1-\rho) \left( \lambda_{3j} - \frac{h\Phi_{\lambda_{33}}}{2} \right) \right\} \mathcal{S}^*_{j-1} + \left\{ \lambda_{4j} - \frac{h\Phi_{\lambda_{43}}}{2} \right\} \left\{ r_2 + \mu^*_{1(j-1)} \right\}$$

$$+ \left\{ \lambda_{3j} - \frac{h\Phi_{\lambda_{33}}}{2} \right\} \left\{ u^*_{1(j-1)} + \eta + \eta_1 + r_2 - \Gamma\xi\alpha \right\},$$

$$\Phi_{\lambda_{44}} = \eta \left\{ \lambda_{4j} - \frac{h\Phi_{\lambda_{43}}}{2} \right\}, \quad \Phi_{\lambda_{54}} = \left\{ \left( \lambda_{5j} - \frac{h\Phi_{\lambda_{53}}}{2} \right) - \left( \lambda_{1j} - \frac{h\Phi_{\lambda_{13}}}{2} \right) \right\} \zeta$$

$$+ \eta \left\{ \lambda_{5j} - \frac{h\Phi_{\lambda_{53}}}{2} \right\},$$

then the algorithm of optimal control problem is concluded in the table as given below.

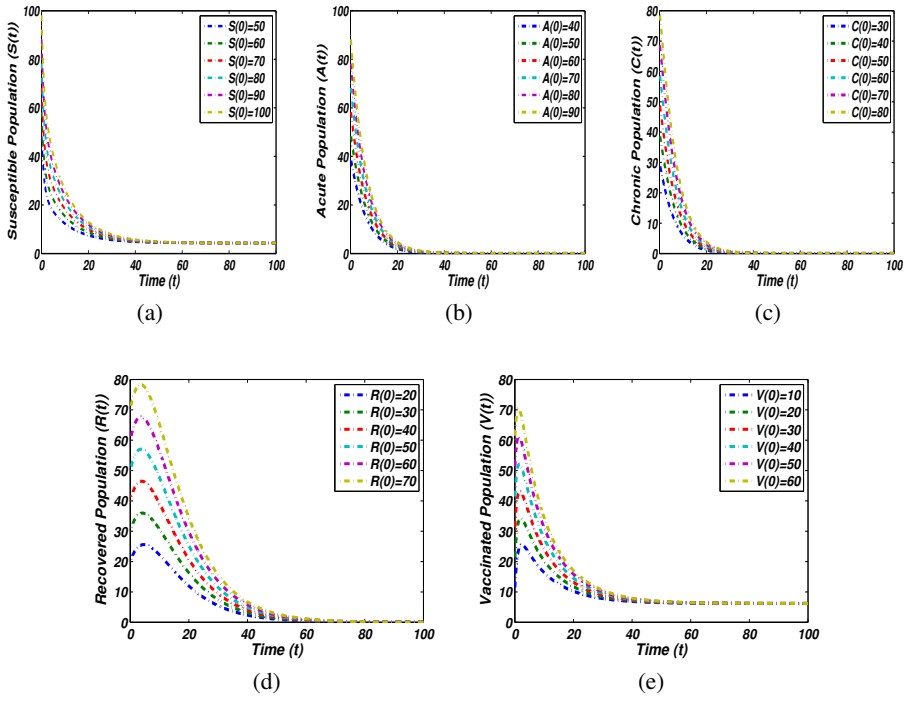

Figure 2: The graphs visualize the solution trajectories for $\mathcal{S}(t)$, $\mathcal{A}(t)$, $\mathcal{C}(t)$, $\mathcal{R}(t)$, and $\mathcal{V}(t)$ of the proposed epidemic problem around disease free state.

---

**Algorithm 2**

---

1: **Input:** Endpoints $t_0$, $t_{\max}$, integer $n$, parametric values, initial conditions

2: **Output:** approximation $\mathcal{S}, \mathcal{A}, \mathcal{C}, \mathcal{R}, \mathcal{V}$ at $(n+1)$ values of $t$

3: **Parameters and Initial Conditions:** Setting the values for model parameters, weights constants, and for the initial sizes of population $\mathcal{S}(0), \mathcal{A}(0), \mathcal{C}(0), \mathcal{R}(0), \mathcal{V}(0)$.

4: **for** $i = 1, \cdots, n$ **do**

5:    **Recursive Formulas** for both control system and without control system:

$$t_i = t_0 + ih,$$

$$\mathcal{S}_{i+1} = \mathcal{S}_i + \frac{h}{6}\left(\Phi_{\mathcal{S}1} + 2\Phi_{\mathcal{S}2} + 2\Phi_{\mathcal{S}3} + \Phi_{\mathcal{S}4}\right), \mathcal{A}_{i+1} = \mathcal{A}_i + \frac{h}{6}\left(\Phi_{\mathcal{A}1} + 2\Phi_{\mathcal{A}2} + 2\Phi_{\mathcal{A}3} + \Phi_{\mathcal{A}4}\right),$$

$$\mathcal{C}_{i+1} = \mathcal{C}_i + \frac{h}{6}\left(\Phi_{\mathcal{C}1} + 2\Phi_{\mathcal{C}2} + 2\Phi_{\mathcal{C}3} + \Phi_{\mathcal{C}4}\right), \mathcal{R}_{i+1} = \mathcal{R}_i + \frac{h}{6}\left(\Phi_{\mathcal{R}1} + 2\Phi_{\mathcal{R}2} + 2\Phi_{\mathcal{R}3} + \Phi_{\mathcal{R}4}\right),$$

$$\mathcal{V}_{i+1} = \mathcal{V}_i + \frac{h}{6}\left(\Phi_{\mathcal{V}1} + 2\Phi_{\mathcal{V}2} + 2\Phi_{\mathcal{V}3} + \Phi_{\mathcal{V}4}\right),$$

6: **end for**

7: **for** $i = 1, \cdots, n$ **do**

8:    **Recursive Formulas** for adjoint system:

$$\lambda_{1(j-1)} = \lambda_{1j} - \frac{h}{6}\left(\Phi_{\lambda_{11}} + 2\Phi_{\lambda_{12}} + 2\Phi_{\lambda_{13}} + \Phi_{\lambda_{14}}\right), \lambda_{2(j-1)} = \lambda_{2j} - \frac{h}{6}\left(\Phi_{\lambda_{21}} + 2\Phi_{\lambda_{22}} + 2\Phi_{\lambda_{23}} + \Phi_{\lambda_{24}}\right),$$

$$\lambda_{3(j-1)} = \lambda_{3j} - \frac{h}{6}\left(\Phi_{\lambda_{31}} + 2\Phi_{\lambda_{23}} + 2\Phi_{\lambda_{33}} + \Phi_{\lambda_{34}}\right), \lambda_{4(j-1)} = \lambda_{4j} - \frac{h}{6}\left(\Phi_{\lambda_{41}} + 2\Phi_{\lambda_{42}} + 2\Phi_{\lambda_{43}} + \Phi_{\lambda_{44}}\right),$$

$$\lambda_{5(j-1)} = \lambda_{5j} - \frac{h}{6}\left(\Phi_{\lambda_{51}} + 2\Phi_{\lambda_{52}} + 2\Phi_{\lambda_{53}} + \Phi_{\lambda_{54}}\right),$$

9: **end for**

10: **for** $i = 1, 2, \ldots, n, j = n + 2 - 1$ **do**

11:    **Optimal Control Variable ($\mu_1$)**

$$if \quad \mu_1^*(j) < 0 \quad then \quad \mu_1(j) = 0 \quad else, \quad if \quad 0 < \mu_1^*(j) < 0 \quad then \quad \mu_1(j) = \mu_1^*(j) \quad else \quad \mu_1(j) = 1$$

12: **end for**

13: **for** $i = 1, 2, \ldots, n, j = n + 2 - 1$ **do**

14:    **Optimal Control Variable ($\mu_2$)**

$$if \quad \mu_2^*(j) < 0 \quad then \quad \mu_2(j) = 0 \quad else \quad if \quad 0 < \mu_2^*(j) < 0 \quad then \quad \mu_2(j) = \mu_1^*(j) \quad else \quad \mu_2(j) = 1$$

15: **end for**

16: **Output:** $(t_i, \mathcal{S}_{i+1}, \mathcal{A}_{i+1}, \mathcal{C}_{i+1}, \mathcal{R}_{i+1}, \mathcal{V}_{i+1})$.

---

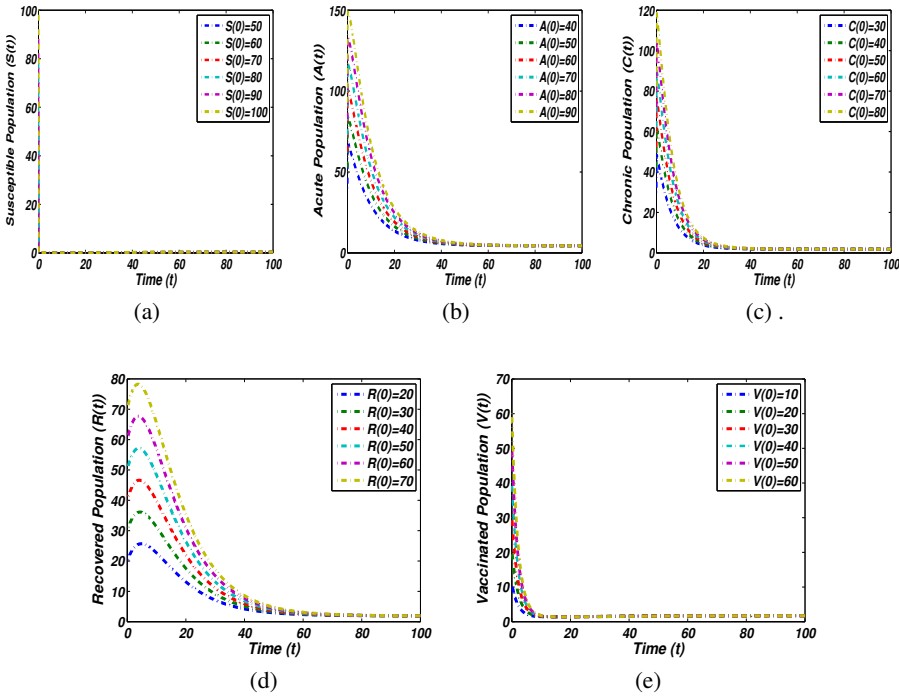

Figure 3: The graphs visualize the solution trajectories for $\mathcal{S}(t)$, $\mathcal{A}(t)$, $\mathcal{C}(t)$, $\mathcal{R}(t)$, and $\mathcal{V}(t)$ of the proposed epidemic problem around the disease endemic state.

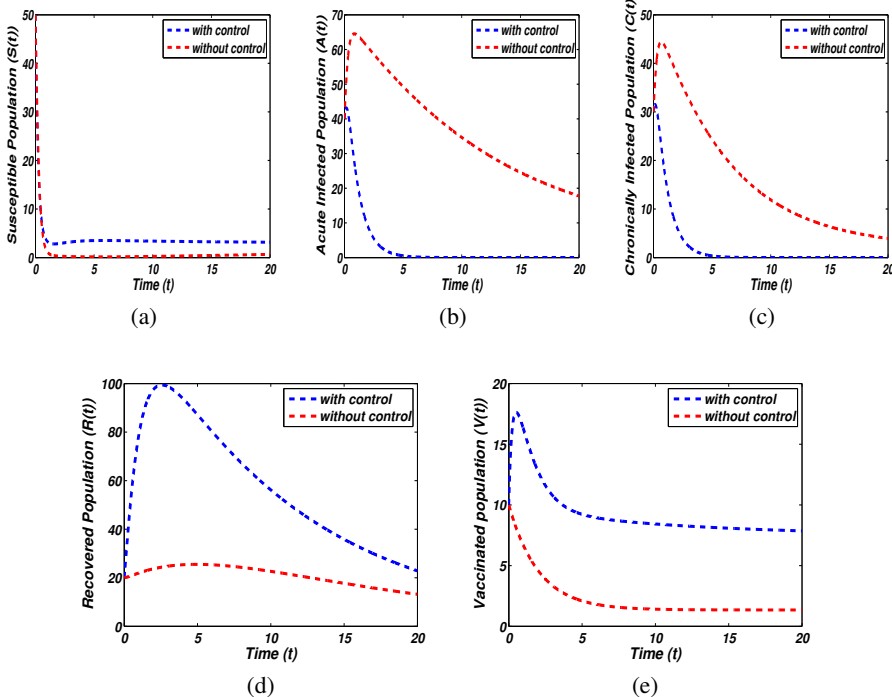

Figure 4: The graphs represent the dynamics of the compartmental population with and without control applications.