# OpenReview forum: "Transmission Dynamics of Hepatitis B: Analysis and Control"
_ICLR.cc/2023/Conference — Submitted to ICLR 2023_

### Official Review · Reviewer_EeVw · 2022-10-22

**Confidence:** 4
**Correctness:** 2
**Technical Novelty And Significance:** 3
**Empirical Novelty And Significance:** Not applicable
**Recommendation:** 3

**Clarity, Quality, Novelty And Reproducibility:**

The paper is not clear in some parts. The proposed mathematical model is not well explained. Also, it is unclear where Lyapunov function theory is applied.

**Strength And Weaknesses:**

**Weaknesses**:

1) First off, I wonder whether this paper would be suitable enough for ICLR? To me, it seems to be more appropriate for a journal in mathematical biology or ...

2) On page 2, Theorem 1:
- $H_3$ is not correct (its last row is incorrect)
- $H_n$ should be written in a more general form, please see the following references:

[1] Süli, E., Numerical solution of ordinary differential equations. Lecture note, University of Oxford, (2013).

[2] F. R. Gantmacher, L. B. Joel, Applications of the Theory of Matrices, Courier Corporation, (2005).

3) The graph presented in Figure 1 is not well explained, and so the modeling of the system (4)-(5) is unclear.

4) In Section 4, the authors have referred to Lyapunov function theory to investigate the global dynamics/stability of the model. However, if I understood correctly, this method is not explicitly employed in this paper (and the stability analysis has been performed using
linearization and Hurwitz criteria). Hence, the authors should make explicit the use of Lyapunov function theory in their work.

5) On page5, Theorem 2 is **incorrect**. The mentioned condition on $R_0$ is not sufficient for asymptotically stability. The proof of Theorem is not correct either. If  $R_0 < 1$  and $a_3,a_4 >0$ , then all the determinants of the Hurwitz matrices cannot necessarily be positive. For instance, under the mentioned conditions, $H_2$ does not always have a positive determinant.



**Summary Of The Paper:**

This paper formulates a mathematical model governing the dynamics of hepatitis B infection. The authors have obtained the threshold parameter to analyze the model equilibria and its stability. They also analyzed a model for hepatitis B virus transmission with optimal control. Some developed algorithms and numerical simulations were carried out  to validate the theoretical results.

**Summary Of The Review:**

Overall, I think the paper is not appropriate enough of ICLR, and there are some incorrect or unsupported claims.

---

### Official Review · Reviewer_FwjM · 2022-10-24

**Confidence:** 5
**Clarity, Quality, Novelty And Reproducibility:** Please see the detailed comments above.
**Correctness:** 4
**Technical Novelty And Significance:** 3
**Empirical Novelty And Significance:** Not applicable
**Recommendation:** 3

**Strength And Weaknesses:**

This paper is technically sound. To show the effectiveness of the proposed model and algorithm, both theoretical support and empirical evaluations are provided. However, this work can be further improved in the following aspects:

• According to the authors, the motivation of this paper is to study the strategies to forecast and control hepatitis B virus transmission. However, based on the data from CDC and WHO, Hepatitis B is not a major threat to us now. Compared to COVID, its transmission rate is not that high, and it is a vaccine-preventable disease. I am not sure if the study for control hepatitis B virus transmission at this point is essential or necessary.

[1] https://www.cdc.gov/hepatitis/statistics/2020surveillance/hepatitis-b.htm
[2] https://www.who.int/health-topics/hepatitis#tab=tab_1

• I have concerns for the novelty of the proposed model and algorithm. Compartmental epidemics models have been well developed. Besides classical SIS or SIR model, there already be some works which divided infectious stages to two separate stages, also the stability of such models have been studied [3] [4]. Compare to these work, I didn’t see much novelty or improvement form this paper. In my opinion, it may be able to bring limited insights to the optimization and epidemics community.

[3] https://arxiv.org/pdf/2010.12923.pdf
[4] https://papers.ssrn.com/sol3/papers.cfm?abstract_id=3590621

• Figure 1 is not vector graph, it should be updated.
• The writing of this work can be further polished. There exists a lot of grammar issues.
• This paper is not well-organized. Some of the sections are not consistent with the other part. For example, in Section 2.2, it is not clear why this principle is needed in the whole framework. Also, this section is not stated clearly, where the dynamical from and why there will an optimal problem were not clear. Besides, there exist a lot of long paragraphs which are not easy to follow. For instance, the two paragraphs in introduction, and the two paragraphs in numerical simulation.
• In the numerical simulations, how these parameter values are chosen is not clear.
• The format of Table 1 should be updated to meet the requirement of the conference.


**Summary Of The Paper:**

This work considers the transmission and control problem of hepatitis B. To address this issue, a mathematical dynamical system is constructed. Based on the proposed model, the sensitivity analyses for epidemic parameters are implemented. Also, some control methods were studied and analyzed.

**Summary Of The Review:**

In summary, I think this work is not ready to be published yet. I have concerns for both motivation and novelty of the proposed model and algorithms.  Also, the organization of this paper is messy, and there exists some format issues.

---

### Official Review · Reviewer_aznX · 2022-10-24

**Confidence:** 3
**Correctness:** 4
**Technical Novelty And Significance:** 1
**Empirical Novelty And Significance:** Not applicable
**Recommendation:** 1

**Clarity, Quality, Novelty And Reproducibility:**

Clarity & Quality: The paper is mostly about mathetical derivation and proof.

Novelty: The paper claims the novelty of the proposed model, but this is not readily verifiable unless you are an expert in the transmission of Hepatitis B Virus.

Reproducibility: The paper does not provide any source code, hence the numerical analysis would be difficult to reproduce.

**Strength And Weaknesses:**

Strengths:
- The paper derives a complex HBV transmission model using well-developed mathematical tools.
- The paper provide a deep analysis of the proposed model and shows its stability and the existence of a solution

Weaknesses:
- All the modeling and analysis, solution derivation is done for only one disease, HBV, which is far from the topic of interest for ICLR audience.
- The entire analysis of the proposed model is purely mathematical, and the paper does not show the model generalizes well to the real-world data.
- The paper is not self-contained, as the main analysis results are in the appendix.

**Summary Of The Paper:**

This work proposes a new dynamical system for modeling the transmission of Hepatitis B virus (HBV). Based on the classicial SIR (susceptible-infected-recovered) framework, the paper develops a new probability-based HBV transmission model, investigates the sensitivity of the model parameters, discuss the method to eradicate HBV using optimal control theories.

**Summary Of The Review:**

Overall, this paper is proposes a purely mathematical model of HBV transmission without a real-world application, a topic outside the usual scope of ICLR. The paper should be submitted to a more relevant venue such as BMC Public Health or JMIR Health and Surveillance.

---

### Official Review · Reviewer_znhF · 2022-11-01

**Confidence:** 2
**Clarity, Quality, Novelty And Reproducibility:** Well written paper, no code release f…
**Correctness:** 3
**Technical Novelty And Significance:** 3
**Empirical Novelty And Significance:** 2
**Recommendation:** 3

**Strength And Weaknesses:**

This is not a representation learning or ML paper. There will be limited interest in the paper for the ICLR audience. This work should be submitted to and be peer-reviewed in a mathematics journal/conference or a more related venue.

**Summary Of The Paper:**

This paper provides a mathematical compartment based model for Hepatitis B which uses a probability based transmission to be more realistic and consider different disease phases.  It studies the steady state and used optimal control theories to find a control mechanism for the eradication of HBV transmissions, without needing simulations.

**Summary Of The Review:**

Not within the scope of ICLR.

---

### Decision · Program_Chairs · 2023-01-20

**Decision:**

Reject

**Justification For Why Not Higher Score:**

Several concerns (on the model, relevance, novelty etc.) raised by all reviewers.

**Justification For Why Not Lower Score:**

N/A

**Metareview: Summary, Strengths And Weaknesses:**

This submission considers the transmission and control problem of Hepatitis B via a dynamical system is constructed. Sensitivity analyses for epidemic parameters are implemented, and some approaches to control are analyzed.

This paper is technically sound.

The referees found many issues, including that Hepatitis B is not a major threat now, has a lower transmission rate than, e.g., COVID-19, and is vaccine-preventable. Additional concerns about novelty were also raised. The authors are asked to incorporate all of these carefully.